# A Qualitative Study on the Motivators, Barriers and Supports to Participation in a Pediatric Produce Prescription Program in Hawai‘i

**DOI:** 10.3390/ijerph192416682

**Published:** 2022-12-12

**Authors:** Monica Esquivel, Alicia Higa, Andrea Guidry, Cherese Shelton, May Okihiro

**Affiliations:** 1Department of Human Nutrition, Food and Animal Sciences, College of Tropical Agriculture and Human Resources, University of Hawai‘i at Mānoa, Honolulu, HI 96822, USA; 2Elepaio Social Services, Waianae, HI 96792, USA; 3Waianae Coast Comprehensive Health Center, Waianae, HI 96792, USA

**Keywords:** qualitative research, child nutrition, native Hawaiian

## Abstract

Produce prescriptions that provide vouchers to individuals to purchase fresh FVs at a specified retail outlet have the potential to positively impact food security status, diet, and chronic disease risk. However, maximizing program participation is vital to ensuring program success. This research describes motivators, barriers, and support for participation in a child produce prescription program among a population of Native Hawaiian and Other Pacific Islanders, who are underrepresented in this field of research. This qualitative grounded theory study was nestled within a quasi-experimental pilot intervention trial and included semi-structured interviews with pediatric program participants. Twenty-five interviews were conducted, which represented one-third of program participants. The top support factors for program participation included: ease of voucher use, program convenience, health center/pediatrician endorsement and positive communications with farmers’ market vendors. Key motivators for program participation were produce enjoyment, child support, financial support, and positive impacts on family. Three themes emerged consistently as barriers to participation amongst participants, (1) difficult use of vouchers, (2) conflicting schedules, and (3) online market concerns. (4) Conclusions: This research offers insight into policy implications as the number of produce prescription programs has grown. These findings suggest that key program design characteristics can enhance and support program participation.

## 1. Introduction

Clinical and community-based interventions to increase fruit and vegetable (FV) intake, while reducing food insecurity, in children have had limited success in improving diet [1,2]. A promising exception is adult produce prescription (Produce Rx) programs. Produce Rx programs provide vouchers to individuals to purchase fresh FVs at a specified retail outlet [3]. Across the United States nearly 100 new programs were established between 2010 and 2020 [4]. Evidence from US and international bodies of research suggest that these programs have a positive effect on household food insecurity [5,6], improve fresh FV consumption [7], and improve chronic disease management among adults [8,9,10,11]. For example, an intervention at a safety-net clinic that provided food insecure, hypertension patients with a Produce Rx, yielded significant increases (47 to 50%) in daily FV consumption [7]. FV vouchers, as little as $11 per month, have been found to be effective in increasing FV intake and reducing food insecurity in low-income children and adults who participate in the Federally funded Supplemental Nutrition Program for Women, Infants, and Children (WIC) program [12]. The WIC program observed a significant increase in FV purchases and consumption when FV vouchers were introduced [13].

Maximizing participation is an important first step in determining the effect of the FV voucher program for families. For example, in Arizona, a WIC program observed inconsistent redemption rates for WIC FV vouchers, so researchers sought to identify factors that influenced participation, barriers, and attitudes around redemption [14]. The researchers found negative stigma was associated with participation, which lowered redemption rates [14]. In another study, African American WIC participants reported transportation as a barrier to using FV vouchers at farmers markets [15]. Other reported barriers are availability, convenience, transportation, quality, and variety of FV [16,17]. The knowledge gained from these qualitative explorations yielded programmatic changes and innovations to improve redemption, such as mobile farmers markets. Subsequently, it was observed that WIC participants who used vouchers at farmers markets were twice as likely to consume the recommended number of FV servings per day (Odds Ratio = 2.01, 95% Confidence Interval = 1.15–3.5) [13].

### Development of the Keiki Produce Prescription (KPRx) Program

The KPRx program is the result of a collaboration between the Waianae Coast Comprehensive Health Center (WCCHC) Pediatric Department, WCCHC Health Promotion Department, and University of Hawai‘i at Mānoa researchers [18]. KPRx was designed to serve the Wai‘anae Coast of the island of O‘ahu, a low-income, rural community (population 43,609) with a high proportion of Native Hawaiian and Other Pacific Islander residents (37%) [19]. WCCHC is the state’s largest Federally Qualified Community Health Center and the leading healthcare and safety net provider, on the Wai‘anae Coast. WCCHC Health Promotion Department has a strong track record for addressing food access in the Wai‘anae community. WCCHC has operated a weekly community farmers’ market since 2008 which serves between 900 and 1200 customers each week. In a community with low access to fresh foods, the farmers’ market has played a key role in improving food access in the Waianae community. The market was one of the first farmers’ markets in the state to accept Supplemental Nutrition Assistance Program (SNAP) benefits (i.e., food stamp benefits) and provide the double bucks program, which provides a dollar-for-dollar match on local fruits, vegetables, and proteins purchased using SNAP benefits. 

The KPRx feasibility program provided $24 in vouchers to obtain FV each month for three months, but while parents expressed their excitement about and satisfaction with the program, only 23% of participants completed the program [18]. As a result of the feasibility program results, the KPRx program was adapted for pilot testing in 2019. Adaptations included increasing the monthly vouchers to $50 and extending the program to six months. Further adaptations were made in response to the COVID-19 pandemic which included expansion of voucher redemption venues to include a local online food hub, FarmLink Hawai‘i. This expansion provided an opportunity for KPRx participants to receive produce, while adhering to local COVID-19 isolation protocols, as vouchers could be redeemed online and produce delivered to the participant’s home. In the pilot study, KPRx participants had the opportunity to select redemption method (in-person or online) and could switch between methods at any time. Paper vouchers were provided to participants who selected the in-person farmers’ market. Paper vouchers were treated as cash by participating farmers’ market vendors. Farmers’ market vendors were provided with training on the KPRx program, including what items the vouchers could be utilized to purchase. Once trained, market vendors were provided with a sign so that program participants knew which farmers’ market vendors were participating in the KPRx program. Vouchers were provided to participants who selected the online FarmLink option via a monthly credit to their personal online account. Participants could log in to their FarmLink accounts to access the vouchers. In both venues, participants could use their vouchers to purchase any fresh fruits or vegetables they preferred. 

The pilot study aimed to quantify the effect of the KPRx program on household food insecurity status and FV consumption, as well as identify mechanisms to improve program participation. The purpose of this qualitative research was to understand factors that KPRx participants perceive as motivators, barriers, and support for KPRx participation in a population of Native Hawaiian and Other Pacific Islanders, who are underrepresented in this field of research.

## 2. Materials and Methods

This qualitative grounded theory study was nestled within the KPRx pilot intervention trial. Participants were referred to the KPRx pilot study by their WCCHC pediatrician. Pediatricians screened patients for food insecurity using a validated 2-question tool [20]. Patients who screened positive for food insecurity were referred to the KPRx program by the pediatrician via an automated referral mechanism in the patient’s electronic medical record. KPRx research staff received referrals and then contacted patients to verify program eligibility. Participants were eligible for the KPRx program if they were (1) patients of the WCCHC pediatric clinic, (2) two to 18 years of age; (3) screened positive for food insecurity using a validated two-question tool, and/or had overweight or obesity (defined as Body Mass Index for Age ≥ 85th percentile); (4) resident of the Waianae Coast of Oahu (zip code 96792) and (5) English-speaking. Nearly 95% of WCCHC patients are English-speaking, the pilot study was unable to provide translation services, thus only English-speaking patients could be enrolled. We allowed up to two children per household to enroll in the study. 

Consent and assent (children ≥ eight years of age) were obtained prior to participation in KPRx activities, this included consent to be interviewed. Parents/caregivers were invited to participate in the KPRx exit interview once their child had been enrolled in KPRx for at least six months. Parents/caregivers were offered a $25 gift card as a token of appreciation for participation. The research protocol was approved by the Institutional Review Board at the Waianae Coast Comprehensive Health Center (No. 00006399).

The one-on-one interviews were conducted using open-ended questions developed in the feasibility evaluation (Table 1) [18]. These questions were developed to understand the participants’ experiences in navigating the program, their motivation to participate in the program, thoughts on the program’s effectiveness, and satisfaction with the program. The questions were also reviewed by the WCCHC Community Advisory Group, experienced researchers, and Waianae community members.

Interviews were conducted between 2021 and 2022 via phone at time convenient to the participant. Participants completed a brief demographic questionnaire (i.e., child sex, age, ethnicity, household information) at enrollment in the KPRx study. 

Interviews were conducted and recorded by the study principal investigator who had little interaction with participants throughout the duration of the pilot study, to limit bias in responses. The principal investigator has a good understanding of the community and Native Hawaiian and other Pacific Islander culture, having worked in the target community for over 10 years. Interviews lasted between 30 and 60 min. Audio recordings were transcribed verbatim using an audio transcription service. KPRx research staff reviewed transcripts for accuracy. As interviews were conducted, research staff performed reviews of data. Data collection was ceased when saturation was achieved (e.g., no new themes were emerging).

Two trained members of the research team analyzed the interview transcripts using a grounded theory approach [21]. Each researcher independently read transcripts line by line and open coded the transcripts via the Nvivo software Version 1 (QRS software 2020). The two researchers compared codes and discussed any discrepancies until consensus was reached. Final codes were clustered into related categories and grouped into broader themes. The themes were used to understand factors that serve to support KPRx program participation, factors that motivated participation and barriers to program participation. Recommendations for future program offerings were also identified. Frequencies and counts for themes were also tracked to demonstrate prevalence of the cited factors and to underscore the number of interviewees that mentioned the theme. The Principal Investigator reviewed coding, categories, and themes. The final themes and codes were approved by the research team. 

## 3. Results

### 3.1. Sample Characteristics

A total of 95 KPRx study participants were eligible to complete an exit interview (enrolled for at least six months). Interviews were conducted with 25 parents/caregivers, representing 34 children who participated in the KPRx pilot study (nine interviewees had two children enrolled in the KPRx program). All parents/caregivers were women (*n* = 25). The average age of participating children was nine years (ranging from two to 18 years). Ninety-two percent (*n* = 31) of the child participants were of Native Hawaiian or Other Pacific Islander race. Over half of the interviewees received SNAP (food stamp) benefits (*n* = 18) and 68% (*n* = 17) had completed the KPRx program, defined as having redeemed vouchers for all six months of the program. 

### 3.2. Supports for Program Participation

The top support factors for program participation included: (1) ease of voucher use, (2) program convenience, (3) health center/pediatrician endorsement and (4) positive communications with farmers’ market vendors.

#### 3.2.1. Ease of Use

All participants shared that the ease of use of the vouchers supported their program participation, this was mentioned 35 times across interviews (*n* = 25). Specific examples of what made the vouchers easy to use included that they vouchers were treated as cash and were welcomed by all vendors. One participant shared, “Everybody in the marketplace accepted it. And they were happy and everybody treated you, you know, nicely. Just like as if you were giving them money.”

#### 3.2.2. Convenience

The program’s convenience was shared as a support for program participation by 13 participants, 16 times. Specifically, being able to use the vouchers at the community farmers’ market, having options for weekly participation (for example, in-person or online markets), not having to wait in line at the farmers’ market to get the vouchers, and the ability to redeem at several different market vendors were some of the most convenient characteristics. One participant shared, “To me, it’s easy to access, or it’s easy to really just go in, and especially picking up the coupons…It’s not long line because every time I go, it’s already ready. By saying that, they are organized” and another shared: “Being able to just go to the market and pick it up, and then from whatever tent they were in, and then my kids could just go and use it on their own. It was just like money.”

#### 3.2.3. Health Center/Pediatrician Endorsement

Ten of the twenty-five participants mentioned a support for program participation was through endorsement by the health center and the pediatrician specifically, mentioned 11 times. One participant shared, “The doctor recommended us and I was like why not. It helps me get vegetables, so I was like okay, why not, we’ll see what this program is about.” And another shared, “We actually was referred by my daughter’s doctor and that’s what got us involved in participating.”

#### 3.2.4. Positive Communication with Vendors

Participants (*n* = 8) mentioned specifically that the interactions with the vendors supported participation (cited nine times). Most of the comments in this theme were around vendors helping participants to utilize the full voucher benefit. For example, if the voucher value was five dollars and the participant selected four dollars in FV, the vendor would help the participant in identifying what they could get for one dollar to maximize their benefits. One participant shared, “The vendors, they were very helpful. If I didn’t, sometimes it was a specific thing, if it was too high, they would put something else they would add on top of, like a banana. And that’s how, because I couldn’t break the $5 [voucher], they would put on more stuff for me so I could use that certain amount up.”

### 3.3. Motivators for Program Participation

Four key subjects emerged as motivators for program participation. The topics, in order of most frequently stated, were: (1) produce enjoyment, (2) child support, (3) financial support, and (4) positive familial effect.

#### 3.3.1. Enjoyment of Produce

More than half (16 out of 25) interviewees mentioned produce enjoyment 18 times across the interviews. Participants enjoyed the process of purchasing and picking the produce as well as the availability and accessibility of the produce. Interviewees shared that children were excited to purchase FV. For example, a participant stated, “So, I know my son is very enthusiastic, every time I tell him it’s time to go pick up his vouchers, and he likes picking out the fruit and vegetables for us.” Participants enjoyed the availability and accessibility of produce at the farmers market, stating, “… it’s harder to get fruits and veggies with all the prices going up, or even just getting those type[s] of fruits and vegetables is more in the stores. Sometimes it’s limited, too. There’s none at the stores. So it’s easier to get at the market”.

#### 3.3.2. Support for Child

The second frequently cited motivator was the support provided to the child, which was mentioned 17 times by 15 participants. Some participants noted the program was “[a] great way of allowing [my daughter] to try different types of vegetables and fruits”. Parents also perceived increased FV consumption as “support[ing] a healthier lifestyle”. One user stated, “I chose to participate in the program so my son would eat healthier. He would have more options on vegetables, fruits, so he could learn more about the different types of vegetables and fruits that you guys offered in the program.”

One participant was motivated to join the program to support their child’s community engagement. She, and another, mentioned, “I was just kind of curious to see what kind of resources they would have. And what kind of ways they would’ve had the community get engaged, especially with the younger kids.” Another shared, “I thought that it was a good program for the kids, just because it promoted eating healthier and that they got to try all different types of vegetables and fruits that they usually don’t try. And I like how the program, how they got those bags with the stuff inside to make the vegetables. I like those and the recipes that it came with, all the information. I thought that was cute.”

#### 3.3.3. Financial Help

Financial support was the third largest program motivator, mentioned 21 times by 12 people. Some users mentioned that it increased FV affordability, “Anytime we can get some fresh fruits and vegetables at a good cost, that’s free. Right? That’s awesome.” Others mentioned that the help purchasing food freed the financial burden to purchase other household necessities. For example, “It was easy and very helpful, because toward the end of the month, when you don’t have enough food, that helped out a lot. It helped out with just buying the necessities that we need, besides groceries, other things that it could help us with in the household.” One participant said it helped the family provide food with the financial burdens of the COVID-19 pandemic, “I know it would benefit us by helping, because it was kind of tight with the budget, especially dependent on the COVID and all that, and being out of work for certain people in our household. So that help food-wise, a whole bunch actually helps.”

#### 3.3.4. Positive Family Effect

Twelve participants mentioned the positive family effect of the program, which came up 17 times across interviews. One person noted that they as a parent were also able to enjoy the benefits produce, saying, “I was able to eat with them [my family]”. Another person mentioned that it gave the family something to do together, “ … going to the farmer’s market actually gave us a reason to go out on Saturday and be a part of the community.”

### 3.4. Barriers to Participation

Three themes emerged consistently as barriers to participation amongst participants. In order of frequency, barriers were the (1) difficult use of vouchers, (2) conflicting schedules, and (3) online market concerns.

#### 3.4.1. Difficult Use of Vouchers

Thirteen participants mentioned some difficulty in using the vouchers 30 times across the interviews. Factors that made the vouchers difficult to use as perceived by participants ranged. Some participants thought that the expiration dates complicated their use. Participants said, “I’d have to buy things that I wouldn’t want to buy before the thing [voucher] would expired.”. Another shared, “I was getting confused with the expiration date on the bottom. I thought when I could use it was the expiration date.”

Although positive vendor communication did help, some participants found the need to use the whole voucher made the vouchers difficult to use. For example, “because the vouchers were in whole dollar increments, it made it a little difficult.” Other concerns included wanting to use the vouchers for products that were not FV.

#### 3.4.2. Conflicting Schedule

A conflicting schedule between participants and the market was mentioned by 9 participants and referenced 11 times. One participant stated, “just organizing our own schedule as a family was the issue”. Other participants mentioned that Saturdays were normally filled with obligations that could not be done during the work week. Some had other familial obligations, “Sometimes I really wanted to get the vegetables but I had to choose. Do I go to my son’s game or to this family event or do I pick up my vegetables.” Another shared, “It was just harder for me if we had appointments on Saturdays and we had to drive all the way to town, there was no way for us to be able to go to the farmer’s market. I mean, I don’t know if it is really, I mean, most people have time I am supposing on Saturday mornings, but for working parents during the week, it is just, Saturday is the only day for them to get stuff done.”

#### 3.4.3. Online Market Concerns

The final and third most noted barrier to participation was online market concerns. Online market concerns were referenced 7 times by 6 participants. Common concerns with the online market included not getting to make produce selections, in terms of preference and quality selection, also noting that the online option had a smaller selection of produce, “there was not as much to choose from as if you went to the farmer’s market…” One person also mentioned, “But one sort of onion, which I know they have the purple and the regular onions, but we like the purple ones. They just dropped off like a gang load of the regular ones, And… just being able to choose what we like. I wouldn’t have chosen the oranges. I would’ve probably went with some green onions. So just being able to choose.”

### 3.5. Recommendations for Future Programs

Participants offered some specific recommendations to improve the program for the future, these included adjusting the voucher values and increasing advertisement for the program.

#### 3.5.1. Voucher Values

Most participants found the value of the vouchers value sufficient. One participant voiced, “I think that was a perfect amount. I’d be greedy if I asked for more”. Though sufficient, a consistent recommendation was to provide the voucher in smaller increments. Smaller values suggestions were referenced 5 times by 4 participants. Vouchers were available in five and ten-dollar increments. One participant mentioned failing to use the full voucher value, stating, “I guess when something was half of a dollar, then we had to give them the whole dollar. Or they didn’t give change back, that’s why. So we kind of lost out on that part of it.” Other participants also suggested smaller increment amounts, “maybe just being able to split it change-wise.”

#### 3.5.2. Advertisement for Program

More advertisement on the program was referenced three times throughout participant exit interviews. One participant “never knew it existed”. Another expressed that they would share about the program with friends and family who did not know about it, “No, but I shared it with all my friends, and I think a lot of them would try and see if they could get into the program as well. As I shared with many of my friends, who were going through some difficulties as well with the pandemic and losing their job, they said, “Oh.” They didn’t know they had the program.”

## 4. Discussion

This qualitative research study gathered perspectives regarding supports, motivators, barriers, and recommendations for future program optimization from 25 parent/guardians who had a child enrolled in a Produce Rx program in Hawai‘i. The research findings suggest that the health center and pediatrician endorsement of the program as well as additional program design qualities made the program easy and convenient which inevitably supported program participation. Participants were particularly motivated by the positive effect that the program had both on the child and shared that the benefits were experienced by other family members, even though the program was really designed for just the participating child. This was similar to another pilot study which found that enhanced family bonding and positive experiences for children as a benefit of a Produce Rx program [22].

Barriers to participation included scheduling time to attend the market to redeem the vouchers. Online prescription redemption with home delivery service was designed to address this barrier. However, this alternative did not appeal to many KPRx participants. Participants found the online market unappealing, citing the smaller variety of produce available online and the inability to hand pick the produce as reasons for not favoring this option. Contrary to our findings, previous research has found that delivery services enhanced participation in Produce Rx programs among low-income households in urban settings [22]. In the future, program improvements could be made to overcome other barriers cited by participants. These may include providing vouchers in smaller increments to enable participants to spend the full amount of the voucher as well as clarifying voucher expiration dates which created confusion for participants. Interestingly, the farmers’ market vendor’s knowledge and communication helped to overcome this barrier, as participants shared that the vendors often helped them to maximize the benefits.

Findings from this study suggest that several key program design qualities were beneficial to encouraging program participation. These included leveraging the trusted relationship between the health center, pediatrician, and participants, which is supported in the literature [23]. The well-informed farmers’ market vendors and well-designed voucher pick up process supported participation as well. Past studies identified stigmatization as a barrier to WIC voucher use at farmers market, where the stigmas were often connected to the final payment transaction and interaction with the market vendor or cashier [16,24]. Well trained and accepting cashiers facilitated a pleasant experience among those studies [16,24]. In this study, the vendors familiarity with the program may have reduced this sort of stigmatization. Additionally, the well-designed pick-up process, which included appointment times for pick-ups expedited the process and ensured there was no line for participants to wait in. While this process was initiated to adhere to social distancing guidelines during COVID-19, it actually allowed program staff to anticipate who would be arriving for pick up, which may have also minimized the stigma associated with receiving benefits and participating in program. Notably, this process became the KPRx standard procedure even as COVID-19 restrictions were eased, as staff also found it to be efficient.

Motivating factors were identified in this research, which can inform future recruitment and retention efforts. Parents and caregivers were motivated to support their child both in improving their health through diet and providing their child with the opportunity to try new FVs. This is key, as Native Hawaiian and Other Pacific Islander cultures place high value on family and collective good over individual benefits. While the KPRx program was designed to primarily benefit the child, parents/caregivers expressed the positive effect that the program had on the whole family. Future programs can leverage this motivating factor several ways; (1) expand definition of participant from child to family unit, (2) include outcome assessment and evaluation at the family or household level, and (3) consider expanding benefits to support the family/household unit.

Participants experienced barriers related to scheduling to attend the farmers’ market, despite our program’s partnership with the local online food hub which offered participants the flexibility to select produce and have it delivered free of charge to their home. Several participants expressed the inability to hand pick produce on the online platform as a barrier to utilizing the option. The opportunity to choose and select foods has been cited as a best practice for increasing participant satisfaction and diet quality among other food assistance programs for individuals experiencing food insecurity (i.e., food banks and food pantries) [25,26,27]. Alternative options that allow for participant selection, might include expanding fruit and vegetable voucher redemption to retail grocery stores in the community. While farmers’ markets are the most common venue for Produce Rx programs (48%), nearly one-third of Produce Rx programs rely on retail grocery stores which suggests this could be a viable strategy [4]. Another strategy may include identifying additional farmers’ markets that may be more convenient to parents’ place of work or by offering pop-up markets at various locations in the community (e.g., near schools and parks).

### Strengths and Limitations

This novel study investigated motivators, barriers, and supports to participation in a pediatric Produce Rx program, by female parent/caregivers of children who were enrolled in the program. It is the first to do so in a primarily Native Hawaiian or Pacific Islander population. This qualitative study has addressed and met the 21 criteria of the Standards for Reporting Qualitative Research Guidelines [28]. Limitations of this research include the relatively small sample size, which limited our group’s ability to differentiate results by program participation completion. Another limitation of the study was reliance on a convenience sample of program participants who were willing to complete the interview. This may have led to bias in the responses and subsequently limits generalizability of the results.

## 5. Conclusions

This research has identified key themes related to the KPRx that can inform program optimization and testing. The program design led to well trained staff and vendors who then facilitated an engaging and convenient experience for program participants, ultimately supporting program participation. Despite having two venues to redeem vouchers, the online marketplace was not an adequate solution for some participants, which created a challenge for participants to schedule time to attend the market. The addition of a retail grocery store is a potential solution presented by participants to overcome this challenge and should be explored. Another strategy to be explored is the potential for an electronic system which could reduce challenges related to the paper vouchers. The KPRx program intends to ease financial barriers to accessing healthy food with the goal of improving the health of children. Interviewees reinforced the same motivation for participating. As such, this messaging can be leveraged to support program recruitment. These findings can ultimately lead to program improvements that will enhance program participation and subsequent impact on diet and health, an important next step to long term sustainability of Produce Rx programs.

## Figures and Tables

**Table 1 ijerph-19-16682-t001:** Keiki (child) produce prescription (KPRx) Interview questions.

Interview Questions
1. Tells us what you liked most about the program.
2. Why did you choose to participate in the program?
3. Do you think it was easy or hard to use the vouchers?
4. What made it easy to use the vouchers?
5. What made it hard to use the vouchers?
6. What can we do to make it easier to use the vouchers in the future?
7. Is there anything else you would like to share with us?

## Data Availability

The data presented in this study are available on request from the corresponding author. The data are not publicly available due to privacy concerns related to a small sample size.

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
