# Peer review of "A Qualitative Study on the Motivators, Barriers and Supports to Participation in a Pediatric Produce Prescription Program in Hawai‘i"

_ijerph, 2022, doi:10.3390/ijerph192416682_

Round 1
Reviewer 1 Report
This is well done. The paper would benefit from describing the food retail options (in person and online) in greater detail, as a context to what people experience when they try to redeem their vouchers.
Reviewer 2 Report
Thank you for an interesting study exploring the desirability of F&V vouchers (produce prescription) targeting food insecure families with young children. This study has the potential to provide valuable insights into the design of an intervention with greater likelihood of adoption.
Introduction - I would suggest expanding on the literature base provided to substantiate your statement that produce prescription is a growing intervention, with studies conducted in countries other than the US amongst comparable communities to illustrate international relevance. You might like to review this recent publication and the cited references for this purpose - https://www.ncbi.nlm.nih.gov/pmc/articles/PMC9502043/
It would be useful to include a sub-section within the introduction regarding the development of the KPRx intervention - what is its underpinning aim, how are vendors made aware of the scheme, how are participants made aware, why were farmers markets decided as a purchase point, why $50 vouchers etc. Some of this may be in ref 10, however some context would be beneficial to the reader.
Methods - As a qualitative study I strongly suggest you include the use of the COREQ or SRQR guidelines to ensure you are following best practice with the reporting of your study. For example, these guidelines outline the need to include detail regarding the individuals conducting/analysing the interviews and their epistemological and ontological viewpoints.
There also needs to be some explanation provided as to your use of counts and frequencies of the coded data, given this is not a quantitative study.
Please explain why "English-speaking" was an eligibility criterion? (Line 86)
What demographic data was collected? (Line 101)
How were the interview questions developed? Are these based on questions used in other similar studies? Were the questions pilot tested for understanding?
Results - Line 123 mentions SNAP benefits, but this has not been introduced into the publication - please include this in the introduction to provide context for this result.
You state that 68% of the interviewees completed the program ie: redeemed all vouchers across the 60-month intervention period. What about the other 32% - how long did they stay in the program? It would be interesting for you to unpack interview answers between these 2 cohorts to see if barriers were different, especially what led participants to exit the program early.
Discussion -
You mention that collection of vouchers was altered during COVID to ensure social distancing, and this may have minimised stigma - Has this remained or changed post COVID? (Line 308)
More explanation is needed about the "brick and mortar" grocery stores (Line 332) - is there evidence to suggest that this is a feasible recommendation?
Strengths and limitations of the study need to be included.
Conclusion - Currently this contains recommendations about policy implications which are not directly related to your study findings. Your study conclusion is articulated in lines 338-339 ie: to inform future produce prescription program design to enhance uptake and redemption of vouchers. Perhaps consider a separate section entitled "implications for policy, practice and research" in which you can expand on what's needed across the three areas to support produce prescription programs. The conclusion should then focus on what's needed to specifically enhance the KPRx.
Quotations - in some instances these are in italics - good to be consistent
Spelling and grammar:
Line 25 - .. positive family effects...
Line 134 - .... the vouchers were all mentioned (this is unusual wording - do you mean these factors made it easier for vouchers to be redeemed?)
Line 243 - ... stood ... (this is usual word usage - do you mean 'was'?)
Lines 289-290 - this sentence structure needs rewording
Line 305 - check referencing here (superscript) which is different to referencing through manuscript
Line 328 - individuals experiencing food insecurity (not 'with')
Round 2
Reviewer 1 Report
The changes are comprehensive. This is a good paper.
Reviewer 2 Report
Reviewer comments have been addressed appropriately